# Factor Structure of the GHQ-12 and Their Applicability to Epilepsy Patients for Screening Mental Health Problems

**DOI:** 10.3390/healthcare11152209

**Published:** 2023-08-05

**Authors:** Weixi Kang

**Affiliations:** Department of Brain Sciences, Imperial College London, London W12 0BZ, UK; wk20@imperial.ac.uk

**Keywords:** epilepsy, mental health, GHQ-12

## Abstract

Epilepsy, a severe neurological disorder impacting approximately 50 million individuals worldwide, is associated with a high prevalence of mental health issues. However, existing research has predominantly examined the relationship between epilepsy and depression or anxiety, neglecting other dimensions of mental health as assessed by factor scores from the general health survey (GHQ), such as the GHQ-12. This study aimed to explore how epilepsy affects both general mental health and specific dimensions of mental health. By employing a factor analysis and a predictive normative modeling approach, the study examined 426 epilepsy patients and 39,171 individuals without epilepsy. The findings revealed that epilepsy patients experienced poorer general mental health and specific aspects of mental health. Consequently, this study highlights the validity of GHQ-12 as a measure of mental health problems in epilepsy patients and emphasizes the importance of considering the impact of epilepsy on various dimensions of mental health, rather than focusing solely on depression or anxiety. Clinicians should incorporate these study results into the development of interventions aimed at enhancing mental well-being in epilepsy patients, ultimately leading to improved outcomes.

## 1. Introduction

Epilepsy is a serious neurological condition that affects around 50 million people around the world [1,2]. The incidence range is between 40 and 70 per 100 thousand people per year in high-income countries, with a higher prevalence in children and older adults [3]. Moreover, the incidence rate is even higher in low-income countries, with more than 120 per 100 thousand people per year [4,5]. Nevertheless, in high-income countries, there appears to be a greater occurrence of epilepsy among poor people [4]. The World Health Organization (WHO) states that epilepsy contributes to approximately 0.5 percent of the total annual expenses for all illnesses globally [6]. In the United Kingdom, there are approximately 600,000 people suffering from epilepsy [7].

People with epilepsy have high rates of mental health issues [8], with around 20% of patients experiencing depression or anxiety symptoms, which can lead to adverse outcomes, such as decreased life quality [9,10,11,12,13,14] and poor medical adherence [15,16]. Screening for mental health conditions in epilepsy patients can benefit them from behavioral and/or pharmacological interventions [17,18,19,20,21].

Among mental health issues in epilepsy patients, previous studies extensively focused on depression and anxiety. For instance, a recent meta-analysis of population-based studies found that there is an overall pooled prevalence of 23.1% for depression in epilepsy patients with a significantly increased likelihood of depression in people with epilepsy compared to controls [22]. Additionally, studies on epilepsy patients have also found that co-occurring depression and anxiety disorders are more common in people with epilepsy as compared to the general population [23]. Compared to depression, anxiety is typically seen as the “neglected” psychiatric comorbidity in people with epilepsy [24,25]. Despite the fact that depressive disorders are agreed to be a more common psychiatric comorbidity, recent studies have found that the prevalence of anxiety disorders is comparable to depressive disorders [26] or may even exceed the prevalence of depressive disorders [27]. However, the prevalence of anxiety disorders varies in people with epilepsy, with some recent research estimates from as low as 4.3% [28] to as high as 52.1% [29]. One of the most recent meta-analyses found that the pooled prevalence is 22.9% for depression and 20.2% for anxiety, respectively, in people with epilepsy [30].

The 12-item version of the general health questionnaire (GHQ-12) is a self-administered questionnaire consisting of 12 items, each assessed using a Likert scale. Extensive research in the field has examined the psychometric properties of this questionnaire and has demonstrated its reliability, sensitivity, and specificity as a valid measure of mental health [31,32,33,34,35,36,37,38]. However, there is a debate regarding whether the GHQ-12 should be utilized as a unidimensional scale or as a multidimensional structure. Many studies have provided substantial empirical support for a 3-factor model of the GHQ-12 [39,40,41,42,43,44,45]. This model includes three factors: GHQ-12A, which assesses social dysfunction and anhedonia (6 items); GHQ-12B, which evaluates depression and anxiety (4 items); and GHQ-12C, which measures loss of confidence (2 items). Some proponents of the unidimensional approach argue that the high correlation observed between these factors [31] supports the use of a single overall score. However, recent research utilizing simulated data have demonstrated that imposing a simple structure may artificially inflate correlations between the modeled factors [46]. Therefore, relying solely on these high correlations to establish unidimensionality may result in a self-fulfilling prophecy of oversimplification (i.e., simply using high correlations between factors as justification for using a one-factor solution). Thus, the present study considers both the unitary scale and the 3-factor solution of the GHQ-12.

Previous research has primarily examined the relationship between epilepsy and mental health, specifically focusing on depression and anxiety; however, much less is known about how epilepsy is related to general and dimensions of mental health. The objective is to explore how epilepsy impacts overall mental health and its various dimensions. There are two main hypotheses: First, there are three factors in the GHQ-12 scale, including GHQ-12A, GHQ-12B, and GHQ-12C. Second, the study anticipates that individuals with epilepsy will experience poorer general mental health and various aspects of mental health.

## 2. Methods

### 2.1. Data

This research utilized data from Understanding Society: the United Kingdom Household Longitudinal Study (UKHLS), a study that has been gathering yearly data from a representative sample of UK households since 1991, previously known as The British Household Panel Study (BHPS). The data analyzed in this study were obtained from Wave 1, conducted from 2009 to 2010 [47]. Among the participants included in the analysis after removing missing values, 426 individuals reported having epilepsy, while 39,597 participants reported not being clinically diagnosed with epilepsy. Detailed descriptive statistics can be found in Table 1.

### 2.2. Measures

#### 2.2.1. Epilepsy

Utilizing self-reported epilepsy proves to be a reliable method for identifying epilepsy (e.g., [48]). Participants were asked whether a doctor or other health professional had ever diagnosed them with epilepsy, and they responded with a simple “Yes” or “No” to indicate their condition.

#### 2.2.2. Mental Health

Mental health was assessed using the GHQ-12, a 12-item questionnaire often used to screen for general (non-psychiatric) mental health problems in primary care settings. The Likert method of scoring was employed, with response options ranging from 0 (“Not at all”) to 3 (“Much more than usual”) [49]. To capture overall mental health, a summary score was calculated by summing the responses across all 12 items. A higher score indicated poorer mental health. However, for the factor analysis conducted in this study, the GHQ-12 items were scored on a scale of 1 (“Not at all”) to 4 (“Much more than usual”) before the factor analysis.

#### 2.2.3. Demographic Controls

Demographic controls in the model include age, sex, monthly net income, highest educational qualification, legal marital status, and residence.

### 2.3. Analysis

#### 2.3.1. Factor Model

A factor analysis was conducted using MATLAB 2018a software to examine the factor structure of the GHQ-12 questionnaire in this study, which is a statistical technique used in research and data analysis to test and validate the underlying structure of a set of observed variables. The analysis involved an oblique rotation technique with a pre-specified factor 3. The GHQ-12 questionnaire was expected to have three factors: GHQ-12A, GHQ-12B, and GHQ-12C. A higher score indicates a greater presence of mental health issues.

#### 2.3.2. Linear Model

Initially, four general linear models were trained using demographic variables from health controls as predictors and GHQ-12, GHQ-12A, GHQ-12B, and GHQ-12C as the variables to be predicted. Residuals from these models were normally distributed. In the next stage, the four trained models were applied to data from individuals diagnosed with epilepsy in order to estimate the scores they would have obtained if they did not have epilepsy. Lastly, one-sample t-tests were conducted to compare the actual scores of participants with epilepsy to the predicted scores.

## 3. Results

Consistent with previous studies, the three factors obtained from the factor analysis (chi-squared = 2344.61, *p* < 0.001) included GHQ-12A (social dysfunction and anhedonia; 6 items), GHQ-12B (depression and anxiety; 4 items), and GHQ-12C (loss of confidence; 2 items). The loadings of these items can be found in Table 2.

Training general linear models on healthy controls revealed a main effect of age (F(1, 39,164) = 14.59, *p* < 0.001), sex (F(1, 39,164) = 245.89, *p* < 0.001), monthly income (F(1, 39,164) = 23.21, *p* < 0.001), highest educational qualification (F(1, 39,164) = 22.08, *p* < 0.001), legal marital status (F(1, 39,164) = 23.21, *p* < 0.001), and residence (F(1, 39164) = 35.39, *p* < 0.001) on GHQ-12 summary score. Similarly, there was a main effect of age (F(1, 39,164) = 562.28, *p* < 0.001), sex (F(1, 39,164) = 105.25, *p* < 0.001), monthly income (F(1, 39,164) = 23.58, *p* < 0.001), highest educational qualification (F(1, 39,164) = 82.68, *p* < 0.001), legal marital status (F(1, 39,164) = 23.21, *p* < 0.001), and residence (F(1, 39,164) = 9.87, *p* < 0.01) on GHQ-12A (social dysfunction and anhedonia). Moreover, there was a main effect of age (F(1, 39,164) = 147.07, *p* < 0.001), sex (F(1, 39,164) = 257.62, *p* < 0.001), legal marital status (F(1, 39,164) = 40.92, *p* < 0.001), and residence (F(1, 39,164) = 47.5, *p* < 0.001) on GHQ-12B (depression and anxiety). However, the main effect of monthly income and highest educational qualification was not significant. Finally, there was a main effect of sex (F(1, 39,164) = 152.03, *p* < 0.001), monthly income (F(1, 39,164) = 108.89, *p* < 0.001), highest educational qualification (F(1, 39,164) = 55.20, *p* < 0.001), legal marital status (F(1, 39,164) = 234.8, *p* < 0.001), and residence (F(1, 39,164) = 14.21, *p* < 0.001) on GHQ-12C (loss of confidence). However, the main effect of age was not significant. 

Most importantly, the current study found that epilepsy patients have worse overall mental health as indicated by the GHQ-12 summary score (t(425) = 6.63, *p* < 0.001, Cohen’s d = 0.38, 95% C.I. [0.27, 0.50]), GHQ-12A (t(425) = 5.62, *p* < 0.001, Cohen’s d = 0.34, 95% C.I. [0.22, 0.46]), GHQ-12B (t(425) = 5.41, *p* < 0.001, Cohen’s d = 0.29, 95% C.I. [0.18, 0.39]), GHQ-12C (t(425) = 6.12, *p* < 0.001, Cohen’s d = 0.34, 95% C.I. [0.23, 0.45]). The mean and standard error of predicted and actual standardized scores can be found in Figure 1.

## 4. Discussion

The primary objective of this study was to examine and contrast the disparities in overall mental well-being and specific aspects of mental health between individuals diagnosed with epilepsy and those without epilepsy. To achieve this, a predictive normative modeling approach was employed, which effectively controlled for demographic variables. The main findings were that epilepsy patients are characterized by more general health issues, social dysfunction, and anhedonia as well as depression and anxiety and loss of confidence.

In this study, the factor analysis resulted in the identification of three factors, namely GHQ-12A, GHQ-12B, and GHQ-12C. This three-factor structure observed in our study aligns closely with previous research that also identified three factors within the GHQ-12 measure [39,40,41,42,43,44,45]. Furthermore, as indicated in Table 2, the factor loadings obtained in our study were notably high.

Importantly, the main findings in the current study were that epilepsy patients have poorer general mental health when compared to the scores that would be expected given their demographics, which is largely consistent with the literature regarding the general mental health problems in epilepsy patients [23,50,51,52,53]. The finding that epilepsy patients have social dysfunction and anhedonia problems is also consistent with previous studies that found that epilepsy is related to anhedonia via pathophysiological pathways [54,55,56] and in adolescents [57]. Moreover, the finding that epilepsy patients have worse depression and anxiety problems is largely consistent with previous research [23,24,25,26,27,28,29,30]. Finally, the finding regarding loss of confidence is quite novel, although previous studies have only looked at the confidence of epilepsy knowledge [58,59] and management [60].

There are several possible pathophysiological pathways that can possibly explain the association between epilepsy and mental health [61]. Particularly, the amygdala plays an important role in this relationship because it is part of the corticolimbic circuit for producing fear and anxiety symptoms [62]. In patients with temporal lobe epilepsy, ictal fear is typically associated with epileptic discharges from the mesial temporal lobe area [10]. Evidence from magnetic resonance imaging has also found the link between amygdala atrophy and seizure focus on people with ictal fear [63]. Studies on mouse models of temporal lobe epilepsy also revealed an anhedonic phenotype and inhibiting the glutamine synthetase in the central nucleus of the amygdala induces anhedonic behavior and recurrent seizures [55]. Furthermore, there is a suggestion that the neurotransmitter gamma-aminobutyric acid (GABA) might be involved in both epilepsy and depression and anxiety. This connection is demonstrated by the shared properties of GABAergic antiepileptic drugs, which exhibit both anticonvulsant effects and anxiolytic properties [64]. Consequently, GABA could have a significant role in the underlying mechanisms of epilepsy, depression, and anxiety.

The results from the current study suggest that mental health problems can represent a complicating clinical factor in improving the health of epilepsy patients. Comorbid mental health problems can cause adverse clinical outcomes, which may in turn translate to higher health care use and cost. Patients with mental health problems are more likely to encounter epilepsy-specific hospitalizations, epilepsy-specific ED visits, and epilepsy-specific outpatient visits, which may indicate that comorbid psychiatric conditions may be a contributing factor to epilepsy. As a result, the identification and management of epilepsy patients with mental health are important. Comorbid mental health problems can sometimes be overlooked, particularly when they are less severe (i.e., do not cause marked disability). Furthermore, if epilepsy patients with mental health issues go undetected and untreated, it can lead to consequences specific to epilepsy and psychiatric conditions [21]. These outcomes may arise due to neurologists’ lack of awareness regarding suitable screening tools and effective medications for treating mental health conditions. To address this, a more comprehensive approach to care is recommended, involving a multidisciplinary team within epilepsy clinics [8,21].

While this study possesses notable strengths, there are several limitations that need to be acknowledged. First, the reliance on self-reported measures in this study introduces the possibility of self-reporting bias. To address this concern, future research should incorporate more objective measures, such as clinical assessments. Second, the cross-sectional design employed in this study is inadequate for establishing causality, particularly considering the bidirectional association between epilepsy and mental health (e.g., [65]). Therefore, it is recommended that future longitudinal investigations be conducted to further explore the bi-directional association between epilepsy and mental health. Finally, the focus of this study was limited to individuals with epilepsy within the United Kingdom, thus potentially restricting the generalizability of the findings to people living in other countries. 

## 5. Conclusions

Taken together, the current study found that both general mental health and the dimensions of mental health are negatively affected by epilepsy. This study implies that there is a need to consider how dimensions of mental health are affected by epilepsy rather than solely focusing on depression or anxiety problems in patients with epilepsy. Factors with the GHQ-12 may indicate sub-areas for interventions. Clinicians should utilize findings from the current study to develop interventions that improve mental health in epilepsy patients, which can then lead to better outcomes.

## Figures and Tables

**Figure 1 healthcare-11-02209-f001:**
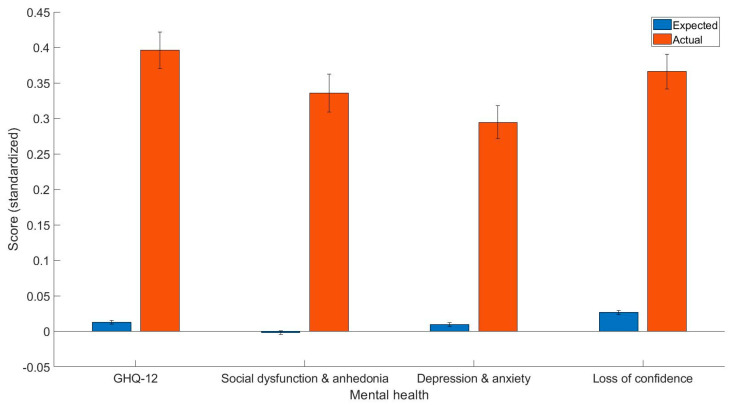
The expected and predicted GHQ-12 summary scores, GHQ-12A (social dysfunction and anhedonia), GHQ-12B (depression and anxiety), and GHQ-12C (loss of confidence). Y axis = standardized scores.

**Table 1 healthcare-11-02209-t001:** The demographic characteristics of the studied sample. The demographic characteristics of these groups of participants are similar.

	Healthy Controls		Epilepsy Patients	
	Mean	S.D.	Mean	S.D.
Age	45.85 (15–99)	17.99	43.31 (16–84)	15.84
Monthly net income	1248.77	1351.45	1046.10	835.41
	N	%	N	%
**Sex**				
Male	21,974	56.10	178	41.78
Female	17,197	43.90	248	58.22
**Highest educational qualification**				
College	11,296	28.84	83	19.48
Below college	27,875	71.16	27,875	80.52
**Legal marital status**				
Single	19,171	48.94	238	55.87
Married	20,000	51.06	188	44.13
**Residence**				
Urban	30,597	78.11	326	76.53
Rural	8574	21.89	100	23.47

**Table 2 healthcare-11-02209-t002:** The factor loadings for the three-factor structure of the GHQ-12. Bold values represent that the item in that row loads the heaviest on the corresponding factor in the column.

GHQ-12 Items	GHQ-12A (Social Dysfunction and Anhedonia; 6 Items)	GHQ-12B (Depression and Anxiety; 4 Items)	GHQ-12C (Loss of Confidence; 2 Items)
Concentration	**0.57**	0.19	−0.11
Loss of sleep	0.01	**0.68**	0.012
Playing a useful role	**0.61**	−0.17	0.13
Constantly under strain	**0.74**	−0.13	−0.02
Problem overcoming difficulties	−0.03	**0.86**	−0.08
Unhappy or depressed	0.08	**0.50**	0.20
Losing confidence	**0.57**	0.23	−0.12
Believe worthless	**0.69**	−0.05	0.04
General happiness	0.01	**0.53**	0.34
Capable of making decisions	0.01	0.17	**0.72**
Ability to face problems	0.10	−0.01	**0.73**
Enjoy day-to-day activities	**0.49**	0.12	0.12

## Data Availability

Publicly available datasets were analyzed in this study. These data can be found here: https://www.understandingsociety.ac.uk.

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
