# Peer review of "Factor Structure of the GHQ-12 and Their Applicability to Epilepsy Patients for Screening Mental Health Problems"

_healthcare, 2023, doi:10.3390/healthcare11152209_

Round 1

Reviewer 1 Report

Overall, this is a well-written manuscript with clinical significance. Early on, I suggest that you consider  the use of a total score to screen for mental health risk with use of sub-sclaes for identification of symptom clusters fo targeted treatment.

Lines 66-67. Explain your point re: oversimplification in more detail.

Line 81; data were-- data = plual.

Lines 94-100- Unclear here re: scoring for factor analysis.

Table 2-- CFA factor structure looks robust.

Your results validate te prior factor structure. Emphasize that .

Results comfirm poorer mental helh for those with epilepsy. OK- already in lit but you add confirmation. So say so.

Discussion; Helpful discussion of brian activity and epilepsy to help explain findings. I suggest moving this p to background section re: your scientific premise. Then refer back to this lit. in discussion.

.Say more re; clinical signifiacnce Re; measure may be helpful to screen for mental health problems and then factors/sub-scales may indicate areas to target fo treatment.

Line 215-- What hypothesis? Specify.

Line 219. Clarify language.

Overall, well done. Good luck with polishing the submission.

overall, well written. Just needs a final review by cope editor to make any final corrections once revised.

Author Response

Dear Reviewer, 

I thank you for reviewing this manuscript. Here is my response with all changes highlighted in yellow: 

Lines 66-67. Explain your point re: oversimplification in more detail.

Done, I have added more details. 

Line 81; data were-- data = plual.

Done. 

Lines 94-100- Unclear here re: scoring for factor analysis.

I have clarified this point. 

Table 2-- CFA factor structure looks robust.

Thanks. 

Your results validate te prior factor structure. Emphasize that .

I have emphasized these in the discussion section.

Results comfirm poorer mental helh for those with epilepsy. OK- already in lit but you add confirmation. So say so.

I have emphasized these in the discussion section.

.Say more re; clinical signifiacnce Re; measure may be helpful to screen for mental health problems and then factors/sub-scales may indicate areas to target fo treatment.

Done.

Line 215-- What hypothesis? Specify.

Done. 

Line 219. Clarify language.

Done. 

Reviewer 2 Report

Dear Author,

First, I would like to extend my congratulations on the completion of your compelling study. Having carefully reviewed the submitted manuscript, I would like to offer some constructive suggestions to enhance the reader's comprehension of the information presented and to elevate the overall quality of the text.

Line 70: The reference to Frontiers in Psychiatry (without citation) appears unclear. Could you please specify which study the stated aims refer to?

Table 1: For improved clarity, could you provide more detailed information on each variable, particularly the age range (from what age to what age)? Additionally, could you clarify whether the reported monthly income is before or after taxes? Furthermore, kindly specify the categories into which divorced and widowed respondents were classified. Finally, a clear definition of "urban residence" and "rural residence" would be beneficial. In the comments section for Table 1, kindly indicate whether the two groups of respondents are equivalent or not.

It would be valuable to include a justification for selecting these particular predictors. Could you also outline the hypotheses regarding the respondents' income or residence?

Line 99: Could you please elaborate on the rationale behind transforming the data in this manner?

Section 2.2.2: It would be advantageous to provide information on the distribution of the individual items comprising the GHQ-12 (means, standard deviation, skewness, kurtosis, etc.).

Line 106: Please verify whether this study indeed employed a CFA (Confirmatory Factor Analysis); the results presented seem to stem from an exploratory factor analysis. In case it is a CFA, it is imperative to present the tested model and the values of the absolute and incremental indices (e.g., RMSEA, SRMR, CFI, NFI, GFI), and the chi-squared test value should not be omitted. Concerning the exploratory factor analysis, kindly specify the chosen method (e.g., principal components method) and provide the values of KMO and Bartlett's test. Furthermore, including the scree plot when determining the number of factors is desirable.

Additionally, please indicate the method used to handle missing values.

Table 2: To enhance clarity, arrange the values in ascending or descending order based on size. Moreover, please specify on which data the factor analysis was performed – the control group only, the epilepsy patients only, or both groups together.

Section 2.3.2: Kindly clarify whether this is a generalized linear model or a general linear model. Additionally, providing information on alternative approaches considered and explaining the advantages of the chosen solution would be beneficial. Furthermore, it is essential to demonstrate that the input variables meet the requirements for the application of the chosen approach, including variable type, normality, and other relevant criteria.

Section 3: Please explicitly state the obtained results, rather than assuming they are already presented in lines 126-141.

Figure 1: Please include an explanation of the meaning of the Y-axis for the reader's better understanding.

In Section 2.3, please furnish more specific information about the training of the data.

In Section 4, kindly revisit the output and interpret the obtained results thoroughly.

Lines 162-168: The summarized results pertain to the exploratory factor analysis, not the confirmatory factor analysis.

Lines 180-193: It seems these details are unrelated to the results presented. Please clarify or remove this information accordingly.

Line 221: The claims made are not substantiated by the data, and the study does not validate the GHQ-12. Please rephrase the statement to reflect the study's actual findings.

Please take into consideration both formal and graphical editing of the text. Additionally, ensure that the references conform to the publisher's requirements, which may be reviewed by others.

Wishing you the best of luck with your revisions.

Author Response

Dear Reviewer, 

I thank you for reviewing this manuscript and providing constructive feedback. Here is my response with all changes highlighted in yellow in the manuscript. 

Line 70: The reference to Frontiers in Psychiatry (without citation) appears unclear. Could you please specify which study the stated aims refer to?

Sorry this is a mistake and I have deleted it. 

Table 1: For improved clarity, could you provide more detailed information on each variable, particularly the age range (from what age to what age)? Additionally, could you clarify whether the reported monthly income is before or after taxes? Furthermore, kindly specify the categories into which divorced and widowed respondents were classified. Finally, a clear definition of "urban residence" and "rural residence" would be beneficial. In the comments section for Table 1, kindly indicate whether the two groups of respondents are equivalent or not.

I have presented information regarding the age range. In addition, I changed the wording monthly income to monthly net income so it reflects that income is after tax. Divorced and widowed participants belong to single. I have also added what the classification of residence is based on. The demographic characteristics of participants are very similar. 

It would be valuable to include a justification for selecting these particular predictors. Could you also outline the hypotheses regarding the respondents' income or residence?

I have outlined the hypothesis regarding the respondent's income and residence. 

Line 99: Could you please elaborate on the rationale behind transforming the data in this manner?

Originally I thought that factor analysis cannot deal with zero values but I was wrong. Results before transformation and after transformation are the same. 

Section 2.2.2: It would be advantageous to provide information on the distribution of the individual items comprising the GHQ-12 (means, standard deviation, skewness, kurtosis, etc.).

I have standardized all these values before further analysis so the mean would be 0, std would be 1, skewness is 0, and kurtosis is also 0. 

Line 106: Please verify whether this study indeed employed a CFA (Confirmatory Factor Analysis); the results presented seem to stem from an exploratory factor analysis. In case it is a CFA, it is imperative to present the tested model and the values of the absolute and incremental indices (e.g., RMSEA, SRMR, CFI, NFI, GFI), and the chi-squared test value should not be omitted. Concerning the exploratory factor analysis, kindly specify the chosen method (e.g., principal components method) and provide the values of KMO and Bartlett's test. Furthermore, including the scree plot when determining the number of factors is desirable.

I have added that it is a case of CFA as I prespecified the factor number as 3. Also, the model fit has been indicated. 

Additionally, please indicate the method used to handle missing values.

I removed people who have missing values and mentioned this in the manuscript.

Table 2: please specify on which data the factor analysis was performed – the control group only, the epilepsy patients only, or both groups together.

The factor analysis was performed on both groups.

Section 2.3.2: Kindly clarify whether this is a generalized linear model or a general linear model. Additionally, providing information on alternative approaches considered and explaining the advantages of the chosen solution would be beneficial. Furthermore, it is essential to demonstrate that the input variables meet the requirements for the application of the chosen approach, including variable type, normality, and other relevant criteria.

I used general linear models and have clarified this. I found that residuals are normally distributed which indicates that these models are valid.

Section 3: Please explicitly state the obtained results, rather than assuming they are already presented in lines 126-141.

I have clarified this point.

Figure 1: Please include an explanation of the meaning of the Y-axis for the reader's better understanding.

Done. 

In Section 2.3, please furnish more specific information about the training of the data.

I have provided more details. 

In Section 4, kindly revisit the output and interpret the obtained results thoroughly.

I have added some descriptive words in each sub-section so the readers can better understand what each result corresponds to. 

Lines 162-168: The summarized results pertain to the exploratory factor analysis, not the confirmatory factor analysis.

Sorry for this confusion, as I changed in the method section, a pre-specified 3 factor was determined thus it would be CFA. I reported model fit so now it should be clear.

Lines 180-193: It seems these details are unrelated to the results presented. Please clarify or remove this information accordingly.

I corrected and used the word "can possibly," which indicates it is speculation rather than a conclusion based on the current study. I think it is important to include them as they are some ideas that can potentially explain the links I found.

Line 221: The claims made are not substantiated by the data, and the study does not validate the GHQ-12. Please rephrase the statement to reflect the study's actual findings.

Done.

Please take into consideration both formal and graphical editing of the text. Additionally, ensure that the references conform to the publisher's requirements, which may be reviewed by others.

The assistant editor will help with this.

Reviewer 3 Report

The manuscript presents a short empirical report concerning the screening of mental health problems in epilepsy patients. Results suggest that this clinical population might show mental health problems other than depression and anxiety and that clinicians must attend to the presence of such problems when implementing interventions aiming at mental health in epilepsy.

As a secondary result, the GHQ-12 (a general mental health screening survey) is explored (confirm the expect three-dimension structure) and, apparently, it seems to be validated as a measure of mental health problems in epilepsy patients. Overall, this study has several strengths (a large sample; a reliable data-analysis approach; interesting results). Can the authors, please, consider my following concerns and suggestions?

Can the GHQ-12 be briefly presented, in the context of the General Health Questionnaire (GHQ)? For instance, providing information concerning its goals (for instance, “screening for general (non-psychotic) mental health problems”) and target population (for instance, primary care patients).

While the authors briefly discussed the possibility of GHQ-12 measuring a unidimensional construct in the introduction, they did not empirically investigate this hypothesis. I do not advocate for a second-order generic factor, as it might be conceptually incompatible with a reflective factor model. However, the authors’ stance on this matter remains unclear to the reader. Surprisingly, the authors reported a total score by summing responses across all items (lines 96-97), seemingly assuming the validity of a global factor for GHQ-12 without testing it in their factor analysis.

It was not clear to me if the authors proceeded with a CFA. The description of the statistical procedures is not clear (indicating MATLAB 2018a is not enough, since the reader does not know the script that was used); the reference to the oblique rotation technique and the presence of cross-loadings (Table 2) suggests that an Exploratory Factor Analysis technique was employed. Please, provide more details (if EFA was used, the extraction method, the amount of explained variance for each dimension, etc).

Does the factor solution obtained correspond to the GHQ-12 factor structures described in the literature?

In the 'normative model,' the authors provided the F statistics and associated p-values to describe the effects of sociodemographic characteristics. However, it would be beneficial to complement this information with an effect measure, such as the standardized beta coefficients or similar metrics. The inclusion of effect measures would allow the reader to assess the magnitude of each predictor's impact on the GHQ dimensions more easily. Presenting this information in a table format could enhance readability and make it more user-friendly, while reserving the text for more general comments.

Finally, one limitation of the study is the restricted information provided by the GHQ-12. Although three dimensions were detected, they are known to be highly correlated so it is not clear if with such a screening instrument is it possible to reliably argue that for mental health problems other than depression and anxiety.

I recommend that the authors exercise caution when they state that “this study highlights the validity of GHQ-12 as a measure of mental health problems in epilepsy patients” (lines 14-15) or “This study implies that GHQ-12 is a valid measure of mental health problems in epilepsy patients” (lines 220-221). Indeed, they are not checking the validity of the GHQ-12 to detect mental health problems in epilepsy, since they do not have access to an independent measure of mental health problems in epilepsy patients. So, I suggest the authors moderate their statement and perhaps include the idea that GHQ-12 has the potential to be sensitive to mental health problems in epilepsy patients.

Minor:

Title: Why “factor structures” (why using the plural)?

Line 70 - Please, omit the reference to Frontiers in Psychiatry

Lines 90-91: How many alternatives do have for the “diagnostic” question? Please, provide the alternatives (or indicate how many alternatives were available to answer).

Lines 163-165: perhaps it is not necessary to repeat the description of the nature of the factors since it was described immediately before (lines 15-161)

Author Response

Dear Reviewer, 

I thank you for reviewing this manuscript. Here is my response with all changes highlighted in yellow: 

Can the GHQ-12 be briefly presented, in the context of the General Health Questionnaire (GHQ)? For instance, providing information concerning its goals (for instance, “screening for general (non-psychotic) mental health problems”) and target population (for instance, primary care patients).

Done. 

While the authors briefly discussed the possibility of GHQ-12 measuring a unidimensional construct in the introduction, they did not empirically investigate this hypothesis. I do not advocate for a second-order generic factor, as it might be conceptually incompatible with a reflective factor model. However, the authors’ stance on this matter remains unclear to the reader. Surprisingly, the authors reported a total score by summing responses across all items (lines 96-97), seemingly assuming the validity of a global factor for GHQ-12 without testing it in their factor analysis.

Sorry for this confusion. I modified the manuscript and explained that due to arguments people have, I decided to use both the unitary scale and the 3-factor solution of the GHQ-12.

It was not clear to me if the authors proceeded with a CFA. The description of the statistical procedures is not clear (indicating MATLAB 2018a is not enough, since the reader does not know the script that was used); the reference to the oblique rotation technique and the presence of cross-loadings (Table 2) suggests that an Exploratory Factor Analysis technique was employed. Please, provide more details (if EFA was used, the extraction method, the amount of explained variance for each dimension, etc).

Sorry for this confusion, CFA was actually used with three pre-specified factors. I have acknowledged this in the methods and results section with more details. 

Does the factor solution obtained correspond to the GHQ-12 factor structures described in the literature?

Yes, and I have emphasized this. 

In the 'normative model,' the authors provided the F statistics and associated p-values to describe the effects of sociodemographic characteristics. However, it would be beneficial to complement this information with an effect measure, such as the standardized beta coefficients or similar metrics. The inclusion of effect measures would allow the reader to assess the magnitude of each predictor's impact on the GHQ dimensions more easily. Presenting this information in a table format could enhance readability and make it more user-friendly, while reserving the text for more general comments.

I have added effect size information. 

Finally, one limitation of the study is the restricted information provided by the GHQ-12. Although three dimensions were detected, they are known to be highly correlated so it is not clear if with such a screening instrument is it possible to reliably argue that for mental health problems other than depression and anxiety.

I have adde this as a limitation. 

I recommend that the authors exercise caution when they state that “this study highlights the validity of GHQ-12 as a measure of mental health problems in epilepsy patients” (lines 14-15) or “This study implies that GHQ-12 is a valid measure of mental health problems in epilepsy patients” (lines 220-221). Indeed, they are not checking the validity of the GHQ-12 to detect mental health problems in epilepsy, since they do not have access to an independent measure of mental health problems in epilepsy patients. So, I suggest the authors moderate their statement and perhaps include the idea that GHQ-12 has the potential to be sensitive to mental health problems in epilepsy patients.

Thanks for your suggestion. I have changed now.

Minor:

Title: Why “factor structures” (why using the plural)?

Done. 

Line 70 - Please, omit the reference to Frontiers in Psychiatry

Done. 

Lines 90-91: How many alternatives do have for the “diagnostic” question? Please, provide the alternatives (or indicate how many alternatives were available to answer).

Done. 

Lines 163-165: perhaps it is not necessary to repeat the description of the nature of the factors since it was described immediately before (lines 15-161)\

I believe it should be included as it tells the readers that the current study can replicate the factor structure found in previous studies. 

Reviewer 4 Report

This is the peer-review report of the manuscript entitled Factor Structures of the GHQ-12 and Their Applicability in Epilepsy Patients for Screening Mental Health Problems, submitted to the Healthcare journal. The study sought to validate the factor structure of the 12‐item General Health Questionnaire (GHQ‐12) in individuals reported to have epilepsy within the UK Household Longitudinal Study (UKHLS). The study also examined the epilepsy individuals’ mental health experience compared to the non-epilepsy individuals.

The topic is interesting in the healthcare field. I have a few comments and hope the author will consider them.

Lines 25 – 26, citation needed.

Lines 31 – 35, I suggest the author break this sentence into two and rearrange the second phrase for more explicit meaning. Also, the citations should be placed with each outcome mentioned. For example, … suboptimal adherence to medication regimens [???], decreased life quality [???], reduced life quality [???], poorer educational attainment [???], increased risks of unemployment [???], and increased risk of suicide [???].

Lines 35 – 37, in contrast with the previous sentence, I suggest moving the citation to the end of the sentence.

Lines 39 – 42, this sentence is wordy and unclear. Suggest revising for a concise narrative.

Line 49, “the prevalence of anxiety disorders is variable” – Did the author mean “varies”?

Line 53, GHQ-12 needed all the words spelled out since this is the first time the text introduces the abbreviation.  

Lines 54 – 56, I suggest moving the citations to the end of the sentence.

Lines 68 – 69, I suggest the author revise the transition sentence to emphasize the literature gap identified in the above paragraph.

Lines 69 – 70, please explain “in this manuscript submission to Frontiers in Psychiatry.”

Lines 72 – 74, I suggest rephrasing. The sentence is lengthy but incomplete (missing verb.)

Lines 79, please spell out the words of the abbreviation UK as this is the first time it is introduced in the text.

Line 86, I suggest replacing “healthy controls and epilepsy patients” with “the studied sample.” Also, suggest moving Table 1 to the Results Section.

Line 94 – 100, this paragraph is very confusing. Suggest revising. Does “the study” (line 94) refer to the UK Household Longitudinal Study? The survey items must be specified. Also, what procedure addresses the conflict in point scales (3-point versus 4-point)?

Line 106, please briefly explain the A confirmatory factor analysis (CFA) and its purpose.

Lines 113 – 120, this paragraph is very confusing. I suggest the author clearly define the variables, e.g., independent/predictor and dependent/outcome. Also, I would like to ask the author to explain the term “trained” in line 116. Moreover, I do not understand the sentence on lines 117-119. In addition, I suggest the author revise the sentence to clarify the actual scores (scores from the epilepsy patients) and predicted scores (scores from the non-epilepsy individuals).

Table 2, I suggest the author explain the table and add a footnote. The author should indicate which items represent the factor structure in the table. For example, which six items represent social dysfunction & anhedonia? What do the numbers mean? Why were some of the numbers in bold?

Lines 126 – 140, Where were these results presented? Table?

Lines 142 – 147, the result interpretation is ambiguous.

Lines 156 – 161, I don't see how the authors make this statesman. Perhaps, the results section needs a more transparent interpretation.

Finally, I suggest the author add a conclusion section with a concise summary of the study’s findings and practice implications.

The quality of the English Language is fine—only minor editions for some words and transition sentences are needed. 

Author Response

Dear Reviewer, 

I thank you for reviewing this manuscript and providing constructive feedback. I have now addressed your comments with all changes highlighted in yellow: 

Lines 25 – 26, citation needed.

Done. 

Lines 31 – 35, I suggest the author break this sentence into two and rearrange the second phrase for more explicit meaning. Also, the citations should be placed with each outcome mentioned. For example, … suboptimal adherence to medication regimens [???], decreased life quality [???], reduced life quality [???], poorer educational attainment [???], increased risks of unemployment [???], and increased risk of suicide [???].

I have changed the wording of the sentence so now should be clear. 

Lines 35 – 37, in contrast with the previous sentence, I suggest moving the citation to the end of the sentence.

Done. 

Lines 39 – 42, this sentence is wordy and unclear. Suggest revising for a concise narrative.

Done. 

Line 49, “the prevalence of anxiety disorders is variable” – Did the author mean “varies”?

Yes, and I have corrected this. 

Line 53, GHQ-12 needed all the words spelled out since this is the first time the text introduces the abbreviation.  

Done. 

Lines 54 – 56, I suggest moving the citations to the end of the sentence.

Done. 

Lines 68 – 69, I suggest the author revise the transition sentence to emphasize the literature gap identified in the above paragraph.

Done, 

Lines 69 – 70, please explain “in this manuscript submission to Frontiers in Psychiatry.”

I have deleted this. 

Lines 72 – 74, I suggest rephrasing. The sentence is lengthy but incomplete (missing verb.)

Done. 

Lines 79, please spell out the words of the abbreviation UK as this is the first time it is introduced in the text.

Done. 

Line 86, I suggest replacing “healthy controls and epilepsy patients” with “the studied sample.” Also, suggest moving Table 1 to the Results Section.

Done. 

Line 94 – 100, this paragraph is very confusing. Suggest revising. Does “the study” (line 94) refer to the UK Household Longitudinal Study? The survey items must be specified. Also, what procedure addresses the conflict in point scales (3-point versus 4-point)?

I have changed it so it should be more clear. As described, it only involves changing the codings from 0-1-2-3 to 1-2-3-4. They all have four points but one starts with 0. 

Line 106, please briefly explain the A confirmatory factor analysis (CFA) and its purpose.

Done. 

Lines 113 – 120, this paragraph is very confusing. I suggest the author clearly define the variables, e.g., independent/predictor and dependent/outcome. Also, I would like to ask the author to explain the term “trained” in line 116. Moreover, I do not understand the sentence on lines 117-119. In addition, I suggest the author revise the sentence to clarify the actual scores (scores from the epilepsy patients) and predicted scores (scores from the non-epilepsy individuals).

I have changed the wording so it should be more clear. 

Table 2, I suggest the author explain the table and add a footnote. The author should indicate which items represent the factor structure in the table. For example, which six items represent social dysfunction & anhedonia? What do the numbers mean? Why were some of the numbers in bold?

Bold values represent that the item in that row loads the heaviest on the corresponding factor in the column. For instance, concentration loads heavily on GHQ-12A.   

Lines 126 – 140, Where were these results presented? Table?

They are presented in text only.

Lines 156 – 161, I don't see how the authors make this statesman. Perhaps, the results section needs a more transparent interpretation.\

I changed the wording. 

Finally, I suggest the author add a conclusion section with a concise summary of the study’s findings and practice implications.

Done. 

Round 2

Reviewer 2 Report

Dear Author,

Thank you for diligently revising your manuscript. I must acknowledge that you have adequately addressed the majority of the previous comments. However, there are still some fundamental ambiguities that require attention.

You wrote:

“I have standardized all these values before further analysis so the mean would be 0, std would be 1, skewness is 0, and kurtosis is also 0.”

I am unable to comprehend how you arrived at certain values. To rectify this issue, I kindly request that you include the actual values characterizing the distribution of all 12 items comprising the GHQ-12. Specifically, providing their means, standard deviation, skewness, and kurtosis would enhance the clarity of your work.

You wrote:

“I have added that it is a case of CFA as I prespecified the factor number as 3. Also, the model fit has been indicated.”

I appreciate the alternative approach you have taken to determine the number of factors. It is essential, however, to elaborate on why you opted for three factors and describe the standard parameters of the factor analysis to support your decision. Nonetheless, it is crucial to avoid referring to this procedure as a CFA (Confirmatory Factor Analysis), as it is inherently different. I recommend consulting relevant sources to clarify the distinction, e.g.:

Schreiber, J.B.; Nora, A.; Stage, F.K.; Barlow, E.A.; King, J. Reporting Structural Equation Modeling and Confirmatory Factor Analysis Results: A Review, The Journal of Educational Research 2006, 99, 323-338. https://doi.org/10.3200/JOER.99.6.323-338.

Byrne, B.M. Structural Equation Modeling With AMOS: Basic Concepts, Applications, and Programming, Third Edition (3rd ed.). Routledge, 2016. https://doi.org/10.4324/9781315757421.

Gallagher, M. W., & Brown, T. A. (2013). introduction to confirmatory factor analysis and structural equation modeling. In Handbook of quantitative methods for educational research (pp. 287-314). Brill.

Coping with other comments is fine. Thank you for your commitment to improving the manuscript. I wish you the best of luck as you work on the final revisions.

Sincerely,

Author Response

Dear Reviewer, 

Thank you for your comments. Here is my response: 

  1. I think since the summary score is normally distributed, it is fine for running regression analyses.
  2. I have changed "CFA" to "factor analysis." 

Reviewer 4 Report

Thank you for addressing my comments. 

Author Response

Thank you.